# AdvLoRA: Adversarial Low-Rank Adaptation of Vision-Language Models

## Abstract

Vision-Language Models (VLMs) are a significant technique for Artificial General Intelligence (AGI). With the fast growth of AGI, the security problem become one of the most important challenges for VLMs. In this paper, through extensive experiments, we demonstrate the vulnerability of the conventional adaptation methods for VLMs, which may bring significant security risks. In addition, as the size of the VLMs increases, performing conventional adversarial adaptation techniques on VLMs results in high computational costs. To solve these problems, we propose a parameter-efficient Adversarial adaptation method named AdvLoRA by Low-Rank Adaptation. At first, we investigate and reveal the intrinsic low-rank property during the adversarial adaptation for VLMs. Different from LoRA, we improve the efficiency and robustness of adversarial adaptation by designing a novel reparameterizing method based on parameter clustering and parameter alignment. In addition, an adaptive parameter update strategy is proposed to further improve the robustness. By these settings, our proposed AdvLoRA alleviates the model security and high resource waste problems. Extensive experiments demonstrate the effectiveness and efficiency of the AdvLoRA.

## 1 Introduction

Artificial General Intelligence (AGI), which aims to create intelligent agents that can perform as well as or better than humans on a wide range of cognitive tasks, is a promising topic for both research and industrial products Pei et al. (2019); Goertzel (2014). As vision and language are the most important information of intelligence, Vision-Language Models (VLMs) have become a significant technique for achieving AGI Fei et al. (2022); Achiam et al. (2023).

In recent years, the adaptation of VLMs aims to improve the performance on different downstream tasks and has become a hot research topic. However, through extensive experiments, we find the vulnerability of the conventional adaptation methods, e.g., Full Fine-Tuning (FFT) Wang et al. (2017); Wu et al. (2022); Zhang et al. (2022a), Linear Probe (LP), LoRA Hu et al. (2021), Unidapter Lu et al. (2023), and Aurora Wang et al. (2023) for VLMs, which may bring significant security threats in various domains, such as facial recognition Venkatesaramani et al. (2021); Sharif et al. (2016), medical analysis Finlayson et al. (2019); Ma et al. (2021) and autonomous driving Zhang et al. (2022b); Feng et al. (2021). As shown in Figure 1, we conduct adaptation experiments of VLMs on the natural and attacked data of the MSCOCO Lin et al. (2014) and MSR-VTT Xu et al. (2016) datasets. From these experimental results, we find that the average performance drops about 30.98% on the attacked data. To solve this problem, various techniques are proposed against adversarial attacks by data augmentation Volpi et al. (2018); Morris et al. (2020), attack detection Metzen et al. (2016); Liu et al. (2018) and adversarial training Goodfellow et al. (2014); Liu et al. (2020). As the most effective defense strategy, adversarial training enhances the adversarial robustness of VLMs by retraining the model on mined adversarial examples Madry et al. (2018); Szegedy et al. (2013); Pang et al. (2020).

However, as the sizes of VLMs increase, the conventional adversarial training method with full parameter updating to improve the adversarial robustness of VLMs will lead to high computing and storage costsGan et al. (2020). In recent years, Parameter-Efficient Fine-Tuning (PEFT) technology has garnered widespread attention as a novel adaptation paradigm due to its significant success in adapting large-scale pre-trained models. PEFT technologies can adapt VLMs with extremely

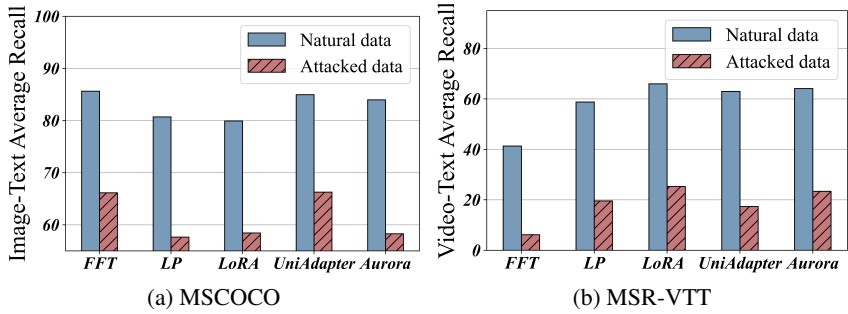

Figure 1: The vulnerability of VLMs adaptation methods on natural data and attacked data.

small additional tunable parameters and achieve comparable or better performance than FFT methods. While PEFT technologies have demonstrated remarkable success in natural scenarios, their application in adversarial attack scenarios remains largely uncharted territory. But simply applying the adversarial training on the conventional adaptation methods will lead to 1) limited defense performance and 2) high computational and storage costs. To verify our points, we visualize the adversarial robustness performance and the tunable parameter number of different adversarial adaptation methods in Figure 2. From the results, we find that the existing adaptation methods such as FFT and UniAdapter will lead to large parameter costs. Besides, LoRA, LP, and Aurora are not robust to adversarial attacks.

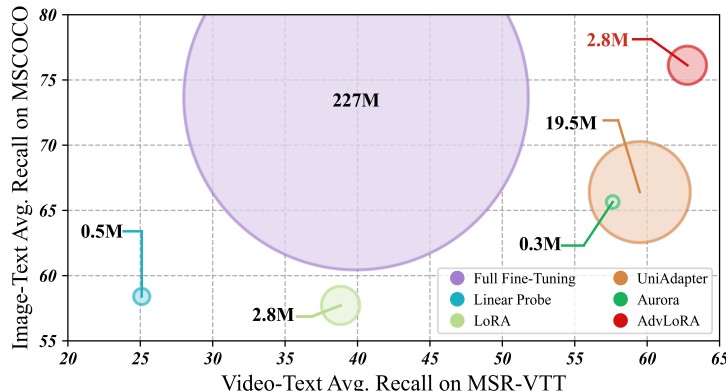

Figure 2: Adversarial robustness and tunable parameter number of adversarial adaptation methods.

To solve these problems, we aim to develop a parameter-efficient adversarial adaptation method termed AdvLoRA to effectively and efficiently improve the robustness of VLMs against attacks. At first, similar to LoRA, the intrinsic low-rank property of adversarial adaptation for VLMs is revealed. Secondly, we improve LoRA with a novel reparameterizing technology. Concretely, we regard the rank of LoRA as the number of cluster centers and utilize the clustering algorithm to reparameterize LoRA from the weight matrices of VLMs. The weight matrices are decoupled into the clustering centers and the clustering distribution matrices. Subsequently, we impose constraints on their product to align with the parameter distribution of the original weight matrix. Moreover, we design an adaptive parameter update strategy to improve the robustness further. Through these settings, we effectively and efficiently facilitate the adversarial adaptation of VLMs. Our designs on low-rank for adversarial adaptation are motivated by the common dense direction theoryAllen-Zhu & Li (2022), which demonstrates that low-rank adaptation in shallow convolutional neural networks are more suitable to effectively enhance their robustness. For the first time, this paper empirically verifies the applicability of this theory to VLMs and introduces a novel clustering-based initialization method for LoRA, facilitating the process of adversarial fine-tuning. The contributions of this paper are summarized as follows.

- We demonstrate the vulnerability of VLMs with different adaptation methods to adversarial attacks via experiments.

- We investigate and reveal the intrinsic low-rank property during the adversarial adaptation for vision-language models.

- We propose a novel parameter-efficient adversarial adaptation method named AdvLoRA with parameter clustering, parameter alignment, and adaptive parameter update.

- We are the first to introduce the adversarial adaptation for vision-language models. Extensive experiments demonstrate the effectiveness and efficiency of our proposed method.

## 2 RELATED WORK

### 2.1 PARAMETER-EFFICIENT TUNING ON VISION-LANGUAGE MODELS

Vision-Language Models (VLMs) have demonstrated success in addressing diverse vision-language downstream tasks, including cross-modal retrievalZeng & Mao (2022); Huang et al. (2023); Geigle et al. (2022) and cross-modal generationRamesh et al. (2021; 2022); Bao et al. (2023); Rombach et al. (2022). However, VLMs may underperform on specific tasks when the data distribution of the task diverges from that of the training data. Consequently, VLMs typically require re-training on task-specific data to effectively adapt to downstream tasks, a process commonly referred to as adaptation or fine-tuning. As the size of VLMs increases, traditional adaptation technologies such as Full Fine-Tuning (FFT) become increasingly inefficient and costlyWang et al. (2017); Wu et al. (2022); Zhang et al. (2022a). Parameter-efficient tuning emerges as a promising solution to alleviate the heavy training and storage costs associated with adapting VLMs.

Recently, inspired by methods from natural language processingHoulsby et al. (2019); Hu et al. (2021); Li & Liang (2021); Liu et al. (2023a); Zhang et al. (2023b); Dettmers et al. (2023) and computer visionRebuffi et al. (2017); Jia et al. (2022); Bahng et al. (2022) domains, some approaches designed for VLMs have been proposed. These approaches aim to adapt frozen VLMs to downstream tasks by introducing extremely small tunable parameters. Despite having fewer tunable parameters, their effects can equal or even exceed that of the full-parameters tuning. These approaches can be broadly categorized into three types: adapter-basedLu et al. (2023); Gao et al. (2023), prompt-basedZhou et al. (2022); Lu et al. (2022); Xing et al. (2022), and LoRA-basedDai et al. (2023); Dou et al. (2023); Hayou et al. (2024); Zhong et al. (2024); Qiang et al. (2024); Liu et al. (2024); Wang et al. (2024); Zhao et al. (2024); Pan et al. (2024). LoRA-based approaches have received considerable attention due to their fewer tunable parameters, no additional input, and no additional inference latency. In this paper, we identify suboptimal initialization in the standard LoRA approach and investigate a clustering-based reparameterization strategy to enhance the robustness of VLMs during adaptation.

### 2.2 ADVERSARIAL ADAPTATION ON VISION-LANGUAGE MODELS

Some researchers have demonstrated that artificial neural networks including Vision-Language Models (VLMs) are vulnerable to human-unrecognized attacks Li et al. (2020); Cai et al. (2023); Zhao et al. (2023). Specifically, adding additional perturbations to input can cause VLMs to make the incorrect decision with high confidence. To improve adversarial robustness on VLMs, most works focus on data augmentation Cai et al. (2023); Wortsman et al. (2022) and adversarial training Gan et al. (2020); Mao et al. (2023). Considered one of the most effective methods, adversarial training can improve the adversarial robustness of VLMs by injecting adversarial inputs into the training procedure through a min-max formulation Madry et al. (2018).

In the early stages of research, some efforts were directed at employing adversarial training techniques to train VLMs from scratchGan et al. (2020). Recently, adversarial adaptation has emerged as a cost-effective strategy for post-pretraining enhancement of adversarial robustnessHendrycks et al. (2019); Liu et al. (2023b); Zhu et al. (2023); Li et al. (2024a;b); Xu et al. (2024); Mao et al. (2023); Yuan et al. (2024); Zhang et al. (2023a). However, the majority of these methods enhance adversarial robustness by updating all the parameters of the pre-trained model through adversarial adaptation, while primarily focusing on the robustness of visual models. A few multi-modal parameter-efficient approaches, such as TeCoAMao et al. (2023), employ prompt tuning for adversarial adaptation, but

they too are limited to classification tasks. In this paper, we utilize a parameter-efficient method based on LoRA to achieve adversarial adaptation for cross-modal tasks. Unlike recent methods like AutoLoRaXu et al. (2024), which aims to solve gradient instability issues by independently extracting natural image features with the LoRA branch, it is essentially not a parameter-efficient approach as it still updates all the parameters. Furthermore, AutoLoRa is exclusively focused on visual models and single-modality tasks.

Our designs to incorporate low-rank methodologies for adversarial adaptation is inspired by the well-established dense direction theory proposed by Allen-Zhu & Li (2022). This theory highlights that integrating low-rank adaptation in shallow convolutional neural networks is particularly effective in bolstering their robustness. Significantly, this paper presents the first empirical validation of this theory within the realm of VLMs. Additionally, it introduces a novel clustering-based initialization method for LoRA, streamlining the adversarial fine-tuning procedure.

## 3 METHOD

In Section 3.1, we first define the cross-modal retrieval. Subsequently, addressing the vulnerability of VLMs to adversarial attacks, we introduce an adversarial training module in Section 3.2 to enhance the model's adversarial robustness. Finally, to mitigate the high cost associated with adversarial training, we present an adaptation module in Section 3.3, which maintains the VLMs' adversarial robustness while reducing the expenses of adversarial training.

### 3.1 TASK DEFINITION

#### 3.1.1 CROSS-MODAL RETRIEVAL

Cross-modal retrieval aims to utilize information from one modality to retrieve semantically relevant information from another. We select cross-modal retrieval as our benchmark task due to its efficacy in assessing the quality of cross-modal representation learning in VLMs. Under adversarial attacks, cross-modal retrieval serves as an effective metric for evaluating whether models can learn robust feature representations.

Taking image-to-text retrieval as an example, given an image $v_i$, its semantic representation $\mathbf{z}_i^v = \mathcal{F}_v(v_i)$ is used to compute the cosine similarity with each textual representation $\mathbf{z}_j^w$ within the text database as follows.

$$\text{sim}(\mathbf{z}_i^v, \mathbf{z}_j^w) = \frac{\mathbf{z}_i^v \cdot \mathbf{z}_j^w}{\|\mathbf{z}_i^v\| \|\mathbf{z}_j^w\|}, \tag{1}$$

where $\mathbf{z}_j^w = \mathcal{F}_w(w_j)$ represents the semantic representation derived from the textual data $w_j$ after feature extraction via the text encoder $\mathcal{F}_w$. Then we select the highest similarity text data as the retrieval results. Under adversarial attacked, robust VLMs could learn semantically invariant feature representations so that they will not be misled by small perturbations.

### 3.2 ADVERSARIAL TRAINING MODULE

Extensive experimentation demonstrated that both VLMs and their variants adapted with PEFT methods are susceptible to adversarial attacks, as illustrated in Figure 1 and the Appendix G. Consequently, in this subsection, we design an adversarial training module to enhance the adversarial robustness of VLMs. We begin by introducing the concept of adversarial attacks, followed by the presentation of adversarial training as an effective defense technology for enhancing adversarial robustness.

#### 3.2.1 ADVERSARIAL ATTACK

Adversarial attacks $\delta$ is a tensor added to the natural image $v$, $v_a = v + \delta$, aiming to fool the model into making the incorrect decision as formulated.

$$v_a = \arg\max_{v_a} \mathcal{L}(v_a, w), \quad \text{s.t.} \quad \|v_a - v\|_p \leq \varepsilon, \tag{2}$$

where $p$ donates the $p$-norm, and $\varepsilon$ donates the restriction value of values, which is often set to be smaller than $8/255$. Thus, the adversarial attacks are imperceptible to humans. In this paper, we focus on adversarial attacks on visual data, as attacks on natural language are readily perceptible to humans. Therefore, it is practically significant and more challenging to make attacks on visual data. Concretely, we utilize PGD Madry et al. (2018) to generate $v_a$ as follows.

$$v_a = \prod \left\{ \text{clip}_\varepsilon (v + \xi \cdot \text{sign} \left( \nabla_v \mathcal{L}(v, w) \right)) \right\}, \tag{3}$$

where $\text{sign}(\nabla_v \mathcal{L}(v, w)))$ denotes the sign value of the back-propagated gradient. Besides, $\xi$ is the step size of each iteration. And $\text{clip}_\varepsilon(x) = \min(x, \varepsilon)$ clips each value of $x$ to be smaller than $\varepsilon$ and return $\varepsilon$ when the value of any dimension exceeds $\varepsilon$. $\prod\{\cdot\}$ denotes the iterative procedure. In this manner, $v_a$ can fool the model to make the incorrect decision. Notably, for video data, we treat it as a collection of images and attack 20% of the frames by randomly sparse sampling Wei et al. (2019).

### 3.2.2 Adversarial Training

Adversarial training technologies refer to retraining the model on attacked data, which can learn semantically invariant features under adversarial attacks. Adversarial training aims to minimize the following objective.

$$\theta = \arg\min_\theta \mathcal{L}(v_a, w), \tag{4}$$

where $\theta$ donates the parameters of the model.

### 3.3 Adaptation Module

Although adversarial training can effectively enhance VLMs' adversarial robustness, it requires updating all parameters based on gradient information, leading to a significant cost overhead. To alleviate this issue, in this subsection, we propose an adaptation module that performs adversarial training on LoRA to reduce the number of tunable parameters, achieving parameter-efficient adversarial adaptation. We first provide a brief introduction to LoRA, followed by the introduction of clustering reparameterization and parameter alignment methods, as well as an adaptive parameter update strategy, to facilitate adversarial adaptation.

### 3.3.1 LoRA

LoRA achieves parameter-efficient adaptation by updating two low-rank matrices attached to the frozen pre-trained weights. Specifically, given the pre-trained weights $\mathbf{W_0} \in \mathbb{R}^{m \times n}$, and the LoRA matrices $\mathbf{A} \in \mathbb{R}^{m \times k}$, $\mathbf{B} \in \mathbb{R}^{k \times n}$, the input $\mathbf{X}^{(l-1)} \in \mathbb{R}^{b \times m}$ is processed through the following computation to obtain the output $\mathbf{X}^{(l)} \in \mathbb{R}^{b \times n}$ as follows.

$$\mathbf{X}^{(l)} = \mathbf{X}^{(l-1)}\mathbf{W_0} + \mathbf{X}^{(l-1)}\mathbf{A}\mathbf{B}, \tag{5}$$

where $k \ll \min(m, n)$. And $\mathbf{A}$ and $\mathbf{B}$ are initialized as follows.

$$\mathbf{A} \sim \mathcal{N}(0, \sigma^2), \quad \mathbf{B} = \mathbf{0}, \tag{6}$$

where $\mathcal{N}$ denotes the Gaussian distribution.

During the adaptation process, $\mathbf{W_0}$ is fixed, while $\mathbf{A}$ and $\mathbf{B}$ are updated via the gradient descent methods. In our proposed model, AdvLoRA, we freeze $\mathbf{W_0}$ and solely update $\mathbf{A}$, $\mathbf{B}$ through adversarial adaptation to achieve adversarial robustness in the model as follows.

$$\theta_{\mathbf{A},\mathbf{B}} = \arg\min_{\theta_{\mathbf{A},\mathbf{B}}} \mathcal{L}(v_a, w). \tag{7}$$

Our model adheres to conventional practice by incorporating LoRA into both the attention modules and feed-forward networks in BLIP.

### 3.3.2 REPARAMETERIZATION AND ADAPTIVE PARAMETER UPDATE

The primary distinction between AdvLoRA and other LoRA-like methods lies in the parameterization process of the matrices $\mathbf{A}$, $\mathbf{B}$. In the original LoRA, a random Gaussian initialization for $\mathbf{A}$ and zero for $\mathbf{B}$, so $\mathbf{AB}$ is zero at the beginning of adaptation. In contrast, our model, AdvLoRA, initially performs clustering on the weight matrix $\mathbf{W_0}$ of the pre-trained model, treating the rank $k$ of LoRA as the number of cluster centers. Specifically, given an weight matrix $\mathbf{W} \in \mathbb{R}^{m \times n}$ and the rank $k$, we first randomly initialize $k$ cluster center $C = \{\mathbf{c_1}, \mathbf{c_2}, \ldots, \mathbf{c_k}\}$. Then, for each column $\mathbf{w_i}$ of $\mathbf{W}$, compute the distances to each cluster center $\mathbf{c_j}$ and assign $\mathbf{w_i}$ to the closest cluster as follows.

$$\text{cluster}_i = \arg\min_{j} \quad \|\mathbf{w_i} - \mathbf{c_j}\|_2. \tag{8}$$

Then update the cluster centers by computing the mean of all data points assigned to each cluster as follows.

$$\mathbf{c_j} = \frac{1}{|\mathbf{S}_j|} \sum_{\mathbf{w_i} \in \mathbf{S}_j} \mathbf{w_i}, \tag{9}$$

where $\mathbf{S}_j$ is the set of columns of $\mathbf{W}$ assigned to cluster $j$. Repeat the above steps until the cluster centers no longer change significantly or a maximum number of iterations is reached. In this manner, we obtain the cluster center embeddings $\mathbf{C} \in \mathbb{R}^{k \times n}$ and the distance assignment matrix $\mathbf{D} \in \mathbb{R}^{m \times k}$, where each element $d_{ij}$ represents the distance between the $\mathbf{w}_i$ and cluster center $\mathbf{c}_j$. The distance assignment matrix $\mathbf{D}$ can be computed using the following formula.

$$\mathbf{d}_{ij} = \|\mathbf{w_i} - \mathbf{c_j}\|_2. \tag{10}$$

And the cluster center representation matrix $\mathbf{C}$ is simply the matrix of cluster centers as follows.

$$\mathbf{C} = [\mathbf{c_1}, \mathbf{c_2}, \ldots, \mathbf{c_k}]. \tag{11}$$

After the parameter clustering, the clustering assignment matrix $\mathbf{D} \in \mathbb{R}^{m \times k}$ and the parameter center $\mathbf{C} \in \mathbb{R}^{k \times n}$ can be represented the $\mathbf{A} \in \mathbb{R}^{m \times k}$ and $\mathbf{B} \in \mathbb{R}^{k \times n}$ in the original LoRA method. By these settings, we provide a better reparameterization of the tunable parameters in LoRA. It separates the parameters into different clusters, which have different functions in the whole network.

After obtaining the matrices $\mathbf{A}$ and $\mathbf{B}$, we further impose constraints on their product $\mathbf{AB}$ to align with the parameter distribution of the original weight matrix $\mathbf{W}_0$ as follows.

$$\min \quad \|\mathbf{W_0} - \mathbf{AB}\|_2. \tag{12}$$

In this manner, we can ensure the initialization of $\mathbf{AB}$ is close to $\mathbf{W_0}$ at the beginning of the training.

During the process of model adversarial adaptation, we design an adaptive update parameter, $\alpha$, to facilitate the model's adaptive learning of robust semantic representations as follows.

$$\mathbf{Y} = \mathbf{XW_0} + \alpha \cdot \mathbf{XAB}. \tag{13}$$

$\alpha$ is a tunable neural network parameter, which can control the adaptation rate during the adversarial adaptation. In summary, we delineate the entire workflow of AdvLoRA in Algorithm 1.

## 4 EXPERIMENT

### 4.1 EXPERIMENTAL SETUP

#### 4.1.1 DATASETS

We comprehensively evaluated our proposed model, AdvLoRA, on two types of retrieval tasks and four commonly used datasets, to demonstrate the superior performance of AdvLoRA on cross-modal understanding tasks, including image-text retrieval: Flickr30K Plummer et al. (2015) and MSCOCO Lin et al. (2014); as well as video-text retrieval: DiDeMo Anne Hendricks et al. (2017) and MSR-VTT Xu et al. (2016), More details can be seen in Appendix A.

---

**Algorithm 1** AdvLoRA WorkFlow on VLMs.

---

0: **Input**: Images: $\boldsymbol{V} = \{v_1, v_2, \ldots, v_n\}$; Texts: $\boldsymbol{W} = \{w_1, w_2, \ldots, w_n\}$; Visual encoder: $\mathcal{F}_v$; Textual encoder: $\mathcal{F}_w$; Pre-trained weight matrix: $\mathbf{W}_0$; LoRA matrix: $\mathbf{A}, \mathbf{B}$; Adaptive parameter: $\alpha$; Restriction value: $\epsilon$; PGD step: $\xi$; Loss function: $\mathcal{L}$.

  **Output**: Representations of $\boldsymbol{V}$ and $\boldsymbol{W}$: $\boldsymbol{Z^v} = \{\mathbf{z}_1^v, \mathbf{z}_2^v, \ldots, \mathbf{z}_n^v\}$, $\boldsymbol{Z^w} = \{\mathbf{z}_1^w, \mathbf{z}_2^w, \ldots, \mathbf{z}_n^w\}$.

  **while** at adversarial fine-tuning stage **do**

    Perform clustering algorithm on $\mathbf{W_0}$ and obtain cluster center representation in Eq. equation 8 and Eq. equation 9;

    Obtain the LoRA matrix $\mathbf{A}, \mathbf{B}$ from cluster center representation and $W_0$ in Eq. equation 10 and Eq. equation 11;

    Impose constraints on $\mathbf{A}$ and $\mathbf{B}$ with SGD algorithm in Eq. equation 12;

    Calculate the loss $l$ using $\boldsymbol{V}, \boldsymbol{W}, \boldsymbol{Y}, \mathcal{F}_v, \mathcal{F}_w$, and the loss function $\mathcal{L}$ in Eq. equation 5.

    Obtain the adversarial attack $\delta$ with $\boldsymbol{V}, \epsilon, k, \xi$ and $l$ in Eq. equation 3.

    Add $\delta$ to original images $\boldsymbol{V}$ to obtain the attacked images $\boldsymbol{V_a}$.

    Update $\mathbf{A}, \mathbf{B}$ via Eq. equation 4.

  **end while**

  Generate robust representations of $\boldsymbol{V}$ and $\boldsymbol{W}$ to downstream tasks with adversarially adapted $\mathbf{A}, \mathbf{B}$ and $\mathcal{F}_v$ $\mathcal{F}_w$.

---

### 4.1.2 BASELINES

We compare AdvLoRA with conventional adaptation methods, which are implemented by BLIP: full fine-tuning (BLIP-FFT), linear probe (BLIP-LP); as well as the PEFT method on BLIP: LoRA(BLIP-LoRA), Aurora, and Uniadapter. See more details in the Appendix B.

### 4.1.3 METRICS

We employ $Recall@k$ as our evaluation metric, where $k$ denotes the number of entries considered within the top $k$ retrieval results. This metric is expressed as a percentage.

### 4.1.4 IMPLEMENTATIONS

Our implementation is based on Salesforce's open-source codebase Li et al. (2022). Following Lu et al. (2023); Wang et al. (2023), we also apply BLIP Li et al. (2022) as our vision-language backbone for all tasks. We also present additional experiments with larger backbones in the Appendix E. We use PyTorch to implement all experiments on the NVIDIA V100 GPU (32GB). We employ PGD-3 Madry et al. (2018) for adversarial adaptation and to assess the model's robustness. Additionally, experiments evaluating the model against a broader range of attack types are detailed in the Appendix D. For the video-text retrieval task, we follow the work of Wei et al. Wei et al. (2019) by adopting an attack strategy that sparsely samples 20% of the video frames. Furthermore, we adopt the setup of BLIP, utilizing a momentum encoder to enhance the retrieval performance of our model. To ensure a fair comparison, the momentum encoder is also applied to the other baseline methods. We use AdamW Loshchilov & Hutter (2018) optimizer with weight decay. The rank of our proposed AdvLoRA is 10. Note that during the fine-tuning process, the parameters of the backbone model are kept frozen. More training details can be seen in the Appendix C.

## 4.2 VULNERABILITY TO ADVERSARIAL ATTACKS

In this section, we conduct adversarial attacks on BLIP and their variants adapted using PEFT methods to investigate their vulnerability to such attacks. Specifically, we perform PGD-3 attacks on the baseline model for two tasks across four datasets and then evaluate their performance under adversarial attacks. Figure 1 provides a simple illustration of the models' vulnerability to adversarial attacks, while Table 1 present detailed data on MSR-VTT dataset. The complete results on other datasets are provided in the Appendix G. Through extensive experimentation, we draw a key conclusion as follows.

BLIP adapted by different methods are highly susceptible to adversarial perturbations. As Table 1 indicate, regardless of whether the method used is full fine-tuning or PEFT, performance degradation of 30.98% is observed. This phenomenon can be attributed to the inability of conventional VLMs and adaptation techniques to effectively learn semantically invariant features from the data.

Table 1: Vulnerability experiment on MSR-VTT. "FFT" and "LP" denoting full fine-tuning and linear probe. "Nat" and "Att" donate natural images and adversarially attacked images. "TR" and "VR" donate text-to-video retrieval and video-to-text retrieval, respectively.

| Method | Tunable Para. | MSR-VTT TR | | | | MSR-VTT VR | | | | |
|---|---|---|---|---|---|---|---|---|---|---|
| | | R@1 | R@5 | R@10 | R@Mean | R@1 | R@5 | R@10 | R@Mean | Mean |
| BLIP+FFT+Nat | 223M | 20.3 | 41.3 | 53.8 | 38.47 | 23.4 | 48.4 | 60.8 | 44.19 | 41.33 |
| BLIP+FFT+Att | | 1.2 | 5.0 | 7.6 | 4.60 | 2.7 | 8.1 | 12.5 | 7.77 | 6.18(-35.15%) |
| BLIP+LP+Nat | 0.5M | 40.3 | 63.2 | 72.0 | 58.50 | 41.8 | 63.7 | 71.6 | 59.03 | 58.77 |
| BLIP+LP+Att | | 7.7 | 16.1 | 20.1 | 14.63 | 14.4 | 26.4 | 32.8 | 24.53 | 19.58(-39.19%) |
| BLIP+LoRA+Nat | 2.8M | 47.2 | 71.4 | 80.5 | 66.36 | 45.8 | 70.7 | 80.3 | 65.60 | 65.98 |
| BLIP+LoRA+Att | | 12.8 | 23.4 | 28.1 | 21.43 | 18.9 | 3.8 | 37.8 | 29.16 | 25.30(-40.68%) |
| UniAdapter+Nat | 19.5M | 42.4 | 68.4 | 77.4 | 62.73 | 42.9 | 68.4 | 78.3 | 63.20 | 62.97 |
| UniAdapter+Att | | 8.3 | 15.4 | 18.9 | 14.20 | 11.6 | 22.6 | 27.2 | 20.47 | 17.33(-45.64%) |
| Aurora+Nat | 0.3M | 45.1 | 69.7 | 79.4 | 64.73 | 44.2 | 68.5 | 77.8 | 63.50 | 64.12 |
| Aurora+Att | | 11.6 | 20.3 | 24.6 | 18.83 | 16.9 | 30.1 | 36.7 | 27.90 | 23.37(-40.75%) |

Table 2: Adversarial experiment on MSCOCO. An asterisk (*) indicates that adversarial adaptation has been performed. The best results are displayed in bold, while the second-best results are underlined.

| Method | Tunable Para. | MSCOCO TR | | | | MSCOCO IR | | | | Mean |
|---|---|---|---|---|---|---|---|---|---|---|
| | | R@1 | R@5 | R@10 | R@Mean | R@1 | R@5 | R@10 | R@Mean | |
| BLIP+FFT+Att | 223M | 53.38 | 75.12 | 82.62 | 70.37 | 42.25 | 67.03 | 76.47 | 61.92 | 66.15 |
| BLIP+FFT*+Att | 223M | _65.42_ | _84.68_ | _89.40_ | _79.83_ | _47.62_ | _73.43_ | _81.35_ | _67.47_ | _73.65_ |
| BLIP+LoRA+Att | 2.8M | 43.20 | 66.20 | 74.80 | 61.40 | 35.85 | 60.40 | 70.16 | 55.47 | 58.44 |
| BLIP+LoRA*+Att | 2.8M | 42.22 | 66.12 | 74.70 | 61.01 | 34.69 | 59.39 | 69.14 | 54.41 | 57.71 |
| BLIP+LP+Att | 0.5M | 43.22 | 65.82 | 74.46 | 61.17 | 34.60 | 58.59 | 68.86 | 54.12 | 57.61 |
| BLIP+LP*+Att | 0.5M | 44.14 | 67.18 | 76.04 | 62.45 | 34.57 | 59.14 | 69.3 | 54.34 | 58.40 |
| UniAdapter+Att | 19.5M | 53.98 | 75.66 | 82.74 | 70.79 | 42.02 | 66.80 | 76.39 | 61.74 | 66.27 |
| UniAdapter*+Att | 19.5M | 50.76 | 76.68 | 85.40 | 70.95 | 39.90 | 67.80 | 77.88 | 61.86 | 66.40 |
| Aurora+Att | 0.3M | 44.56 | 67.04 | 75.00 | 62.20 | 34.98 | 59.34 | 68.75 | 54.36 | 58.28 |
| Aurora*+Att | 0.3M | 54.56 | 77.68 | 84.52 | 72.25 | 40.08 | 60.17 | 75.66 | 60.17 | 65.64 |
| AdvLoRA+Att | 2.8M | 46.76 | 69.18 | 76.72 | 64.22 | 37.00 | 61.25 | 70.76 | 56.34 | 60.28 |
| AdvLoRA*+Att | 2.8M | **67.28** | **87.16** | **92.76** | **82.40** | **49.02** | **75.88** | **84.59** | **69.83** | **76.12** |

## 4.3 PERFORMANCE COMPARISONS

In this section, we conduct a comparative analysis between our proposed AdvLoRA and five baselines across two cross-modal retrieval tasks using four datasets. Specifically, we perform adversarial adaptation based on the PGD-3 attack to all methods and then evaluate their performance under the condition of adversarial attack data and natural data.

Firstly, for image-text retrieval, we conducted experiments on adversarial attacked data for both MSCOCO and Flickr30K, as shown in Table 2 and Appendix F. From these experiments, we draw two important conclusions as follows.

1) After adversarial adaptation, AdvLoRA outperforms all other baselines when faced with adversarial attacks. Notably, on MSCOCO, AdvLoRA surpasses all other PEFT methods by 12.17% and exceeds FFT by 2.47%, while using $\sim 100\times$ fewer tunable parameters than FFT.

2) AdvLoRA demonstrates enhanced adversarial robustness on larger datasets, highlighting the significant potential of PEFT methods in improving model robustness against adversarial attacks. Specifically, on the relatively smaller dataset Flickr30K, the performance of various baselines after adversarial adaptation is comparable and does not show a significant increase in robustness. However, on the larger dataset MSCOCO, FFT achieves considerable adversarial robustness, yet it still lags behind AdvLoRA. These results benefit not only from the design of AdvLoRA in terms of clustering reparameterization and parameter alignment but also indicate that the effectiveness of adversarial adaptation improves with the increase of adaptation data.

Table 3: Adversarial experiment on MSR-VTT. An asterisk (*) indicates that adversarial adaptation has been performed. The best results are displayed in bold, while the second-best results are underlined.

| Method | Tunable Para. | MSR-VTT TR | | | | MSR-VTT VR | | | | |
|---|---|---|---|---|---|---|---|---|---|---|
| | | R@1 | R@5 | R@10 | R@Mean | R@1 | R@5 | R@10 | R@Mean | Mean |
| BLIP+FFT+Att | 223M | 1.2 | 5.0 | 7.6 | 4.60 | 2.7 | 8.1 | 12.5 | 7.77 | 6.18 |
| BLIP+FFT*+Att | 223M | 21.0 | 41.9 | 50.8 | 37.90 | 21.0 | 46.8 | 57.9 | 41.90 | 39.90 |
| BLIP+LoRA+Att | 2.8M | 12.8 | 23.4 | 28.1 | 21.43 | 18.9 | 30.8 | 37.8 | 29.16 | 25.30 |
| BLIP+LoRA*+Att | 2.8M | 21.2 | 43.5 | 52.7 | 39.13 | 21.0 | 42.5 | 52.1 | 38.53 | 38.83 |
| BLIP+LP+Att | 0.5M | 7.7 | 16.1 | 20.1 | 14.63 | 14.4 | 26.4 | 32.8 | 24.53 | 19.58 |
| BLIP+LP*+Att | 0.5M | 14.5 | 26.8 | 33.3 | 24.87 | 15.8 | 26.7 | 33.5 | 25.33 | 25.10 |
| UniAdapter+Att | 19.5M | 8.3 | 15.4 | 18.9 | 14.20 | 11.6 | 22.6 | 27.2 | 20.47 | 17.33 |
| UniAdapter*+Att | 19.5M | 38.6 | 64.0 | 74.5 | 59.03 | 39.2 | 64.9 | 75.8 | 59.97 | 59.50 |
| Aurora+Att | 0.6M | 11.6 | 20.3 | 24.6 | 18.83 | 16.9 | 30.1 | 36.7 | 27.90 | 23.37 |
| Aurora*+Att | 0.6M | 38.1 | 63.6 | 73.5 | 58.40 | 37.0 | 60.8 | 72.7 | 56.83 | 57.62 |
| AdvLoRA+Att | 2.8M | 12.3 | 21.8 | 26.2 | 20.10 | 15.8 | 28.4 | 34.2 | 26.13 | 23.12 |
| AdvLoRA*+Att | 2.8M | **40.4** | **67.4** | **78.6** | **62.13** | **40.5** | **68.4** | **78.4** | **62.43** | **62.28** |

Table 4: Natural experiment with adversarial adaptation on MSCOCO. "Nat" donates natural images. An asterisk (*) indicates that adversarial adaptation has been performed.

| Method | Tunable Para. | MSCOCO TR | | | | MSCOCO IR | | | | |
|---|---|---|---|---|---|---|---|---|---|---|
| | | R@1 | R@5 | R@10 | R@Mean | R@1 | R@5 | R@10 | R@Mean | Mean |
| BLIP+FFT*+Nat | 223M | 57.28 | 78.92 | 86.36 | 74.17 | 48.52 | 75.04 | 83.77 | 69.11 | 71.65 |
| BLIP+LoRA*+Nat | 2.8M | 70.76 | 90.44 | 94.68 | 85.29 | 56.39 | 80.38 | 87.48 | 74.75 | 80.02 |
| BLIP+LP*+Nat | 0.5M | 72.58 | 91.20 | 95.22 | 86.33 | 57.15 | 80.93 | 88.05 | 75.38 | 80.86 |
| UniAdapter*+Nat | 19.5M | 55.62 | 81.24 | 89.02 | 75.29 | 45.06 | 72.99 | 82.61 | 66.89 | 71.09 |
| Aurora*+Nat | 0.3M | 70.92 | 89.39 | 93.94 | 84.74 | 54.38 | 79.38 | 86.88 | 73.54 | 79.15 |
| AdvLoRA*+Nat | 2.8M | 70.58 | 90.42 | 94.54 | 85.18 | 56.36 | 80.35 | 87.29 | 74.67 | 79.92 |

Secondly, for video-text retrieval, we conducted experiments on adversarial attacked data for both MSR-VTT and Didemo datasets, as shown in Table 3 and Appendix F. From these experiments, we draw two conclusions from the image-text retrieval as follows.

1) AdvLoRA achieves excellent adversarial robustness on video data, surpassing all other baselines. In DiDeMo, AdvLoRA slightly outperforms Uniadapter while using 7× fewer parameters. On MSR-VTT, AdvLoRA enhances the model's adversarial robustness by 39.16% and significantly exceeds the other baselines.

2) AdvLoRA demonstrates better adversarial robustness on larger datasets. Specifically, on the relatively smaller dataset DiDeMo, the performance of various baselines after adversarial adaptation is comparable, and the robustness improvement is not significant. However, on the larger dataset MSR-VTT, the Uniadapter method achieves considerable adversarial robustness but is still inferior to AdvLoRA, and it uses 7× more parameters. Such results are attributed to the design of AdvLoRA in terms of clustering reparameterization and parameter alignment. It indicates that the effectiveness of adversarial adaptation improves with the increase of adaptation data.

Thirdly, we conducted experiments on natural data from four datasets, and Table 4 presents the results for MSCOCO. The complete results on other datasets are provided in the Appendix H. From these experiments, we draw a significant conclusion as follows.

Adversarial adaptation can degrade the performance of the model on natural data. For instance, a comparison between Table 4 reveals that, except LP and LoRA, all other models experience a decline in performance after adversarial adaptation. However, the AdvLoRA method still achieves competitive results on MSCOCO. This can be attributed to AdvLoRA's ability to learn semantically invariant feature representations. The reason for the lack of performance degradation in LP and LoRA may be due to their low sensitivity to adversarial adaptation, leading to an ineffective adap-

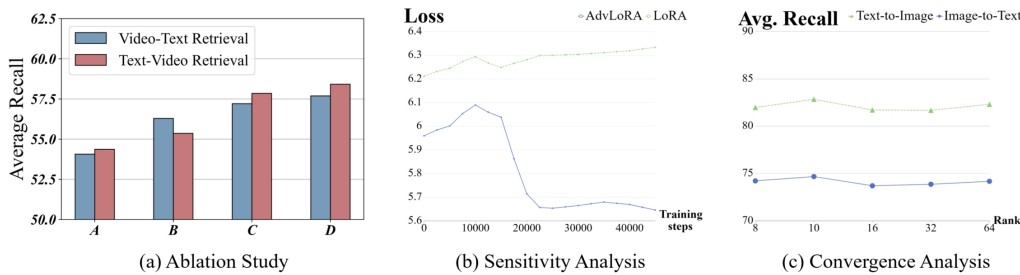

Figure 3: (a) Ablation study. A, B, C, D denotes LoRA, LoRA with parameter clustering, LoRA with parameter clustering and alignment, and LoRA with parameter clustering, alignment and adaptive parameter update (AdvLoRA), respectively. (b) Sensitivity Analysis. (c) Convergence Analysis.

tation process. As shown in Table 2, LP and LoRA do not acquire improved adversarial robustness after adversarial adaptation.

## 4.4 ABLATION STUDY

In this section, we conduct an ablation study on AdvLoRA to demonstrate the effectiveness of the proposed clustering reparameterization, parameter alignment, and adaptive parameter update strategy on Didemo, and the results are presented in Figure 3(a). The model achieves optimal adversarial robustness when these methods are collectively employed.

## 4.5 HYPERPARAMETER SENSITIVITY ANALYSIS

In this section, we conduct a sensitivity analysis on the rank size of AdvLoRA on Flickr30K. We set a series of values for the rank, namely 8, 10, 16, 32, and 64, and the results are presented in Figure 3 (b). AdvLoRA is not sensitive to the rank size, allowing us to select an appropriate rank according to our needs to reduce the cost of adaptation.

## 4.6 LOSS CONVERGENCE ANALYSIS

In this section, we conduct a convergence analysis experiment between AdvLoRA and LoRA on Flickr30K. The results are presented in Figure 3 (c). Analysis of the experimental results we can draw the following conclusion. 1) AdvLoRA demonstrates superior convergence over LoRA in the adversarial adaptation process, achieving a significantly reduced loss level. 2) AdvLoRA accelerates the convergence of adversarial adaptation more effectively than LoRA. These efficiencies and effectiveness can be attributed to the design of clustering reparameterization, parameter alignment, and adaptive parameter update strategy.

## 5 CONCLUSION

In this paper, we aim to alleviate the security risks in the Vision-Language Models (VLMs). First of all, we show the vulnerability of VLMs with various adaptation methods under adversarial attacks via extensive experiments. Besides, as the sizes of VLMs increase, simply applying the conventional adversarial adaptation methods to VLMs easily leads to 1) unpromising adversarial robustness and 2) tremendous parameter and training costs. From these motivations, a novel parameter-efficient adversarial adaptation method named AdvLoRA is proposed with parameter clustering, parameter alignment, and adaptive parameter update. Extensive experiments demonstrate the effectiveness and efficiency of AdvLoRA. This result reveals the intrinsic low-rank property that emerges during the adversarial adaptation process. Our proposed technique, which involves clustering reparameterization and parameter alignment, has been instrumental in facilitating the adaptation process. We have thereby offered a novel perspective for researchers in the field of security within the broader context of AGI. In the future, it is worth further optimizing the memory and computational budget during the adaptation process.

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

## A  DATASETS

- **Flickr30K** Plummer et al. (2015) contains 31,783 images and 158,915 captions totally. Each image is often annotated with 5 captions. Following the split in Uniadapter Lu et al. (2023) and Aurora Wang et al. (2023), we use 1,000 images for testing, another 1,000 for validation, and the rest for training.

- **MSCOCO** Lin et al. (2014) is a large dataset containing 123,287 images and 616,435 captions. Each image is annotated with 5 captions. Following the split in Uniadapter Lu et al. (2023) and Aurora Wang et al. (2023), we use 5,000 images for testing, another 5,000 for validation, and the rest for training.

- **Didemo** Anne Hendricks et al. (2017) contains 10,000 videos and 40,000 annotations. Following Frozen in Time Bain et al. (2021), we concatenate all descriptions corresponding to the same video into a single sentence to conduct actually video-paragraphto retrieval task.

- **MSR-VTT** Xu et al. (2016) is a popular video-text dataset. It contains 10,000 video and 200,000 captions. Following the split in Uniadapter Lu et al. (2023) and Aurora Wang et al. (2023), we use 1,000 videos for testing, another 9,000 for training.

## B  BASELINES

- **BLIP-FFT** is a conventional adaptation technique that enhances the performance of BLIP for specific downstream tasks by retraining and updating full parameters in downstream tasks.

- **BLIP-LP** is an adaptation technique that involves adding and training a linear layer on top of the frozen pre-trained model BLIP to adapt to specific downstream tasks.

- **BLIP-LoRA** is a Parameter-Efficient Fine-Tuning (PEFT) technology that adapts BLIP by introducing low-rank adapters to capture task-specific information, allowing for efficient adaptation to downstream tasks with minimal tunable parameter updates.

- **Uniadapter** Lu et al. (2023) is the first adapter-based PEFT technology for parameter-efficient cross-modal adaptation.

- **Aurora** Wang et al. (2023) is a parameter-efficient cross-modal transfer learning framework that uses mode approximation to generate a minimal set of tunable parameters, achieving lightweight multi-modal adaptation.

## C  HYPERPARAMETER SETTING

We present the hyperparameter setting in Table 5.

## D  PERFORMANCE ON MORE ATTACKS

We conduct more attacks including white-box attack methods (FSGMGoodfellow et al. (2014), PGD-20Madry et al. (2018), BIMKurakin et al. (2022)) and black-box methods (Zeroth Order Optimization (ZOO)Chen et al. (2017), SquareAttack with 500 queries (SA)Andriushchenko et al. (2020)) to further demonstrate the robustness of AdvLoRA in Table 6. Note that AdvLoRA is trained under PGD-3. We observe that our model demonstrates a strong ability to generalize and maintain robustness in the face of various types of attacks.

## E  SCALING TO LARGER BACKBONES

We further demonstrate the effectiveness of our method on larger backbone models on the Flickr30K Dataset in Table 7. Compared to LoRA, our AdvLoRA achieved 2.78%, and 3.25% performance improvements on BLIP-Large, and BLIP-2 (OPT-2.7b), respectively.

Table 5: Hyperparameter setting

| config | Image-text Retrieval | | Video-text Retrieval | |
|---|---|---|---|---|
| | Flickr30K | MSCOCO | Didemo | MSR-VTT |
| optimizer | AdamW | AdamW | AdamW | AdamW |
| learning rate | 1e-5 | 1e-5 | 1e-4 | 1e-4 |
| schedule | cosine decay | cosine decay | cosine decay | cosine decay |
| training batchsize | 16 | 16 | 8 | 8 |
| inference batchsize | 32 | 32 | 8 | 8 |
| frames | - | - | 16 | 16 |
| attack ratio | - | - | 20% | 20% |
| epochs | 5 | 5 | 5 | 5 |
| training input | 384 | 384 | 8*224 | 8*224 |
| inference input | 384 | 384 | 16*224 | 16*224 |
| adversarial type | PGD-3 | PGD-3 | PGD-3 | PGD-3 |
| attack alpha | 1/255 | 1/255 | 1/255 | 1/255 |
| PGD-epsilon | 1/255 | 1/255 | 1/255 | 1/255 |
| rank | 10 | 10 | 10 | 10 |
| adaptive weight | 1e-3 | 1e-3 | 1e-3 | 1e-3 |
| weight norm learning rate | 1e-3 | 1e-3 | 1e-3 | 1e-3 |

Table 6: Additional attack types on the MSCOCO dataset. "TR" and "IR" donate text-to-image retrieval and image-to-text retrieval.

| Method | Attack | TR@Mean | IR@Mean | Mean |
|---|---|---|---|---|
| LoRA | PGD-3 | 61.01 | 54.41 | 57.71 |
| AdvLoRA | PGD-3 | 82.40 | 69.83 | 76.12 |
| AdvLoRA | PGD-20 | 81.65 | 69.13 | 75.39 |
| AdvLoRA | FGSM | 84.44 | 72.21 | 78.32 |
| AdvLoRA | BIM | 83.25 | 69.56 | 76.41 |
| AdvLoRA | SA | 87.40 | 75.83 | 81.62 |
| AdvLoRA | ZOO | 84.17 | 73.83 | 79.00 |

Table 7: Performance on Flickr30K when scaling to larger backbone networks. "TR" and "IR" donate text-to-image retrieval and image-to-text retrieval.

| Backbone | Method | TR@Mean | IR@Mean | Mean |
|---|---|---|---|---|
| BLIP-base | LoRA | 80.40 | 73.24 | 76.82 |
| | AdvLoRA | 82.83 | 74.67 | 78.75 |
| BLIP-large | LoRA | 81.73 | 73.01 | 77.34 |
| | AdvLoRA | 84.74 | 75.56 | 80.15 |
| BLIP-2-opt-2.7b | LoRA | 83.32 | 81.18 | 82.25 |
| | AdvLoRA | 87.34 | 83.66 | 85.50 |

## F  PERFORMANCE COMPARISONS ON MORE DATASETS

We present the performance results on Flickr30K and DiDeMo in Table 8 and Table 9. Specifically, we perform adversarial adaptation based on the PGD-3 attack to all methods and then evaluate their performance under the condition of adversarial attack data and natural data.

Table 8: Adversarial experiment on Flikcr30K. An asterisk (*) indicates that adversarial adaptation has been performed. The best results are displayed in bold, while the second-best results are underlined.

| Method | Tunable Para. | Flickr30K TR | | | | Flickr30K IR | | | | |
|---|---|---|---|---|---|---|---|---|---|---|
| | | R@1 | R@5 | R@10 | R@Mean | R@1 | R@5 | R@10 | R@Mean | Mean |
| BLIP+FFT+Att | 223M | 21.10 | 38.40 | 46.00 | 35.16 | 21.96 | 42.62 | 51.18 | 38.58 | 36.87 |
| BLIP+FFT*+Att | 223M | 64.60 | 84.80 | 87.70 | 79.03 | 55.06 | 79.52 | 84.46 | 73.01 | 76.02 |
| BLIP+LoRA+Att | 2.8M | 67.00 | 81.80 | 84.20 | 77.67 | 58.50 | 77.48 | 82.70 | 72.89 | 75.28 |
| BLIP+LoRA*+Att | 2.8M | 65.60 | **87.10** | 89.50 | 80.40 | 54.62 | 79.92 | 85.18 | 73.24 | 76.82 |
| BLIP+LP+Att | 0.5M | 55.90 | 76.00 | 81.70 | 71.20 | 49.30 | 70.82 | 77.48 | 65.87 | 68.53 |
| BLIP+LP*+Att | 0.5M | 56.10 | 75.70 | 82.70 | 71.50 | 48.14 | 70.50 | 78.18 | 65.61 | 68.55 |
| UniAdapter+Att | 19.5M | 67.20 | 82.50 | 86.50 | 78.73 | 58.26 | 77.26 | 83.30 | 72.94 | 75.84 |
| UniAdapter*+Att | 19.5M | **71.20** | 85.80 | 88.20 | 81.73 | **59.12** | **80.4** | 85.82 | **75.11** | 78.42 |
| Aurora+Att | 0.3M | 65.40 | 80.70 | 84.40 | 76.83 | 56.98 | 76.64 | 82.22 | 71.95 | 74.39 |
| Aurora*+Att | 0.3M | 69.10 | 84.10 | 87.30 | 80.17 | 56.80 | 78.82 | 83.76 | 73.13 | 77.15 |
| AdvLoRA+Att | 2.8M | 66.20 | 82.50 | 85.80 | 78.17 | 57.70 | 77.52 | 83.32 | 72.85 | 75.51 |
| AdvLoRA*+Att | 2.8M | 71.00 | 86.80 | **90.70** | **82.83** | 58.02 | 80.10 | **85.90** | 74.67 | **78.75** |

Table 9: Adversarial experiment on Didemo. An asterisk (*) indicates that adversarial adaptation has been performed. The best results are displayed in bold, while the second-best results are underlined.

| Method | Tunable Para. | Didemo TR | | | | Didemo VR | | | | |
|---|---|---|---|---|---|---|---|---|---|---|
| | | R@1 | R@5 | R@10 | R@Mean | R@1 | R@5 | R@10 | R@Mean | Mean |
| BLIP+FFT+Att | 223M | 12.66 | 26.32 | 35.39 | 24.79 | 14.56 | 31.70 | 40.58 | 28.95 | 26.87 |
| BLIP+FFT*+Att | 223M | 29.71 | 53.04 | 64.30 | 49.08 | 31.21 | 55.63 | 67.40 | 51.41 | 50.25 |
| BLIP+LoRA+Att | 2.8M | 33.20 | 57.43 | 66.70 | 52.44 | 32.70 | 56.73 | 68.10 | 52.51 | 52.48 |
| BLIP+LoRA*+Att | 2.8M | 33.70 | 59.82 | 69.59 | 54.37 | 32.80 | 59.02 | 70.39 | 54.07 | 54.22 |
| BLIP+LP+Att | 0.5M | 23.13 | 45.86 | 53.54 | 40.84 | 26.02 | 47.06 | 57.03 | 43.37 | 42.11 |
| BLIP+LP*+Att | 0.5M | 22.73 | 45.46 | 54.04 | 40.74 | 25.32 | 46.46 | 56.73 | 42.84 | 41.79 |
| UniAdapter+Att | 19.5M | 27.02 | 52.14 | 64.01 | 47.72 | 9.27 | 24.83 | 36.69 | 23.60 | 35.66 |
| UniAdapter*+Att | 19.5M | 36.38 | 63.50 | **73.57** | 57.82 | 35.88 | **64.30** | **73.87** | **58.02** | 57.92 |
| Aurora+Att | 0.6M | 30.31 | 52.94 | 64.11 | 49.12 | 31.21 | 54.74 | 64.61 | 50.19 | 49.65 |
| Aurora*+Att | 0.6M | 35.59 | 61.22 | 72.18 | 56.33 | 36.69 | 62.01 | 71.88 | 56.86 | 56.60 |
| AdvLoRA+Att | 2.8M | 34.40 | 62.11 | 71.39 | 55.97 | 35.19 | 62.81 | 70.99 | 56.33 | 56.15 |
| AdvLoRA*+Att | 2.8M | **37.38** | **64.40** | 73.48 | **58.42** | **36.99** | 63.21 | 72.88 | 57.69 | **58.06** |

# G Vulnerability to Adversarial Attacks

In this section, we present the vulnerability results on Flickr30K, MSCOCO and DiDeMo in Table 10, Table 11 and Table 12. We have the conclusion that BLIP adapted by different methods are highly susceptible to adversarial perturbations, which is similar to that in the main text.

Table 10: Vulnerability experiment on Flickr30K. "FFT" and "LP" denoting full fine-tuning and linear probe. "Nat" and "Att" donate natural images and adversarially attacked images. "TR" and "IR" donate text-to-image retrieval and image-to-text retrieval.

| Method | Tunable Para. | Flickr30K TR | | | | Flickr30K IR | | | | Mean |
|---|---|---|---|---|---|---|---|---|---|---|
| | | R@1 | R@5 | R@10 | R@Mean | R@1 | R@5 | R@10 | R@Mean | |
| BLIP+FFT+Nat | 223M | 72.80 | 90.80 | 95.50 | 86.37 | 63.40 | 86.58 | 92.00 | 80.66 | 83.52 |
| BLIP+FFT+Att | | 21.10 | 38.40 | 46.00 | 35.16 | 21.96 | 42.62 | 51.68 | 38.58 | 36.87(-46.65%) |
| BLIP+LP+Nat | 0.5M | 89.00 | 98.50 | 99.50 | 95.67 | 78.32 | 94.34 | 96.98 | 89.88 | 92.78 |
| BLIP+LP+Att | | 55.90 | 76.00 | 81.70 | 71.20 | 49.30 | 70.82 | 77.48 | 65.87 | 68.54(-24.24%) |
| BLIP+LoRA+Nat | 2.8M | 87.00 | 98.10 | 99.50 | 94.87 | 72.90 | 93.90 | 96.84 | 87.88 | 91.38 |
| BLIP+LoRA+Att | | 71.60 | 92.10 | 95.50 | 86.40 | 60.62 | 85.92 | 91.18 | 79.24 | 82.82(-8.56%) |
| UniAdapter+Nat | 19.5M | 96.70 | 99.70 | 100.00 | 98.80 | 86.18 | 97.34 | 98.82 | 94.11 | 96.46 |
| UniAdapter+Att | | 70.20 | 85.50 | 89.50 | 81.73 | 61.26 | 80.26 | 86.30 | 75.94 | 78.84(-17.62%) |
| Aurora+Nat | 0.3M | 96.70 | 99.80 | 100.00 | 98.83 | 85.76 | 97.24 | 98.72 | 93.91 | 96.37 |
| Aurora+Att | | 69.40 | 84.70 | 88.40 | 80.83 | 60.98 | 80.64 | 86.22 | 75.95 | 78.39(-17.98%) |

Table 11: Vulnerability experiment on MSCOCO. "TR" and "IR" donate text-to-image retrieval and image-to-text retrieval.

| Method | Tunable Para. | MSCOCO TR | | | | MSCOCO IR | | | | Mean |
|---|---|---|---|---|---|---|---|---|---|---|
| | | R@1 | R@5 | R@10 | R@Mean | R@1 | R@5 | R@10 | R@Mean | |
| BLIP+FFT+Nat | 223M | 80.46 | 95.40 | 97.64 | 91.17 | 63.25 | 85.54 | 91.49 | 80.09 | 85.63 |
| BLIP+FFT+Att | | 53.38 | 75.12 | 82.62 | 70.37 | 42.25 | 67.03 | 76.47 | 61.92 | 66.15(-19.48%) |
| BLIP+LP+Nat | 0.5M | 72.30 | 91.10 | 95.22 | 86.21 | 56.96 | 80.75 | 87.85 | 75.19 | 80.70 |
| BLIP+LP+Att | | 43.22 | 65.82 | 74.46 | 61.17 | 34.60 | 58.59 | 68.86 | 54.12 | 57.65(-23.05%) |
| BLIP+LoRA+Nat | 2.8M | 70.50 | 90.28 | 94.58 | 85.12 | 56.39 | 80.36 | 87.45 | 74.73 | 79.93 |
| BLIP+LoRA+Att | | 43.20 | 66.20 | 74.80 | 61.40 | 35.85 | 60.40 | 70.16 | 55.47 | 58.44(-21.49%) |
| UniAdapter+Nat | 19.5M | 79.60 | 94.50 | 97.26 | 90.45 | 62.53 | 84.95 | 90.97 | 79.49 | 84.97 |
| UniAdapter+Att | | 53.98 | 75.66 | 82.74 | 70.79 | 42.02 | 66.80 | 76.39 | 61.74 | 66.27(-18.70%) |
| Aurora+Nat | 0.3M | 78.00 | 93.40 | 96.66 | 89.35 | 61.45 | 83.95 | 90.39 | 78.60 | 83.98 |
| Aurora+Att | | 44.56 | 67.04 | 75.00 | 62.20 | 34.98 | 59.34 | 68.75 | 54.36 | 58.28(-25.69%) |

Table 12: Vulnerability experiment on Didemo. "TR" and "VR" donate text-to-video retrieval and video-to-text retrieval, respectively.

| Method | Tunable Para. | Didemo TR | | | | Didemo VR | | | | |
| --- | --- | --- | --- | --- | --- | --- | --- | --- | --- | --- |
| | | R@1 | R@5 | R@10 | R@Mean | R@1 | R@5 | R@10 | R@Mean | Mean |
| BLIP+FFT+Nat | 223M | 30.51 | 55.63 | 66.40 | 50.85 | 32.80 | 58.52 | 68.49 | 53.27 | 52.06 |
| BLIP+FFT+Att | | 12.66 | 26.32 | 35.39 | 24.79 | 14.56 | 31.70 | 40.58 | 28.95 | 26.87(-25.19%) |
| BLIP+LP+Nat | 0.5M | 25.32 | 44.77 | 53.24 | 41.11 | 26.82 | 50.25 | 58.42 | 45.16 | 43.14 |
| BLIP+LP+Att | | 23.13 | 45.86 | 53.54 | 40.84 | 26.02 | 47.06 | 57.03 | 43.37 | 42.11(-1.03%) |
| BLIP+LoRA+Nat | 2.8M | 36.79 | 63.21 | 72.28 | 57.43 | 34.10 | 62.41 | 73.08 | 56.53 | 56.98 |
| BLIP+LoRA+Att | | 33.20 | 57.43 | 66.70 | 52.44 | 32.70 | 56.73 | 68.10 | 52.51 | 52.48(-4.51%) |
| UniAdapter+Nat | 19.5M | 32.80 | 60.02 | 71.19 | 54.67 | 9.97 | 28.32 | 40.38 | 26.22 | 40.45 |
| UniAdapter+Att | | 27.02 | 52.14 | 64.01 | 47.72 | 9.27 | 24.83 | 36.69 | 23.60 | 35.66(-4.79%) |
| Aurora+Nat | 0.3M | 35.59 | 63.61 | 73.08 | 57.43 | 37.49 | 63.01 | 72.68 | 57.73 | 57.58 |
| Aurora+Att | | 30.31 | 52.94 | 64.11 | 49.12 | 31.21 | 54.74 | 64.61 | 50.19 | 49.66(-7.92%) |

# H PERFORMANCE ON NATURAL DATA

In this section, we present the performance results of natural data on Flickr30K, DiDeMo and MSR-VTT in Table 13, Table 14 and Table 15. We have the following conclusion similar to that in the main text.

Adversarial adaptation can degrade the performance of the model on natural data. However, the AdvLoRA method still achieves competitive results on these datasets. This can be attributed to AdvLoRA's ability to learn semantically invariant feature representations. The reason for the lack of performance degradation in LP and LoRA may be due to their low sensitivity to adversarial adaptation, leading to an ineffective adaptation process.

Table 13: Natural experiment with adversarial adaptation on Flickr30K. "Nat" donates natural images. An asterisk (*) indicates that adversarial adaptation has been performed. The best results are displayed in bold, while the second-best results are underlined.

| Method | Tunable Para. | Flickr30K TR | | | | Flickr30K IR | | | | Mean |
|---|---|---|---|---|---|---|---|---|---|---|
| | | R@1 | R@5 | R@10 | R@Mean | R@1 | R@5 | R@10 | R@Mean | |
| BLIP+FFT+Nat | 223M | 72.80 | 90.80 | 95.50 | 86.37 | 63.40 | 86.58 | 92.00 | 80.66 | 83.51 |
| BLIP+LoRA+Nat | 2.8M | **96.90** | **99.90** | **100.00** | **98.93** | **86.72** | **97.78** | 98.82 | **94.44** | **96.69** |
| BLIP+LB+Nat | 0.5M | 89.00 | 98.50 | 99.50 | 95.67 | 78.32 | 94.34 | 96.98 | 89.88 | 92.77 |
| UniAdapter+Nat | 19.5M | 96.70 | 99.70 | **100.00** | 98.80 | 86.18 | 97.34 | 98.82 | 94.11 | 96.46 |
| Aurora+Nat | 0.3M | 96.70 | 99.80 | **100.00** | 98.83 | 85.76 | 97.24 | 98.72 | 93.91 | 96.37 |
| AdvLoRA+Nat | 2.8M | 96.00 | 99.70 | **100.00** | 98.57 | 85.68 | 97.00 | 98.64 | 93.77 | 96.17 |

Table 14: Natural experiment with adversarial adaptation on Didemo. "Nat" donates natural videoes. An asterisk (*) indicates that adversarial adaptation has been performed. The best results are displayed in bold, while the second-best results are underlined.

| Method | Tunable Para. | Didemo TR | | | | Didemo VR | | | | Mean |
|---|---|---|---|---|---|---|---|---|---|---|
| | | R@1 | R@5 | R@10 | R@Mean | R@1 | R@5 | R@10 | R@Mean | |
| BLIP+FFT+Nat | 223M | 30.51 | 55.63 | 66.40 | 50.85 | 32.80 | 58.52 | 68.49 | 53.27 | 52.06 |
| BLIP+LoRA+Nat | 2.8M | **36.79** | 63.21 | 72.28 | **57.43** | 34.10 | 62.41 | **73.08** | 56.53 | 56.98 |
| BLIP+LB+Nat | 0.5M | 25.32 | 44.77 | 53.24 | 41.11 | 26.82 | 50.25 | 58.42 | 45.16 | 43.14 |
| UniAdapter+Nat | 19.5M | 32.80 | 60.02 | 71.19 | 54.67 | 9.97 | 28.32 | 40.38 | 26.22 | 40.45 |
| Aurora+Nat | 0.6M | 35.59 | **63.61** | **73.08** | **57.43** | **37.49** | **63.01** | 72.68 | **57.73** | **57.58** |
| AdvLoRA+Nat | 2.8M | 32.10 | 60.72 | 69.39 | 54.07 | 35.29 | 59.82 | 71.18 | 55.43 | 54.75 |

Table 15: Natural experiment with adversarial adaptation on MSR-VTT. "Nat" donates natural images. An asterisk (*) indicates that adversarial adaptation has been performed. The best results are displayed in bold, while the second-best results are underlined.

| Method | Tunable Para. | MSR-VTT TR | | | | MSR-VTT VR | | | | Mean |
|---|---|---|---|---|---|---|---|---|---|---|
| | | R@1 | R@5 | R@10 | R@Mean | R@1 | R@5 | R@10 | R@Mean | |
| BLIP+FFT+Nat | 223M | 20.3 | 41.3 | 53.8 | 38.47 | 23.4 | 48.4 | 60.8 | 44.20 | 41.33 |
| BLIP+LoRA+Nat | 2.8M | **47.2** | 71.4 | 80.5 | 66.36 | 45.8 | 70.7 | **80.3** | 65.60 | 65.98 |
| BLIP+LB+Nat | 0.5M | 40.3 | 63.2 | 72.0 | 58.50 | 41.8 | 63.7 | 71.6 | 59.03 | 58.77 |
| UniAdapter+Nat | 19.5M | 42.4 | 68.4 | 77.4 | 62.73 | 42.9 | 68.4 | 78.3 | 63.20 | 62.97 |
| Aurora+Nat | 0.6M | 45.1 | 69.7 | 79.4 | 64.73 | 44.2 | 68.5 | 77.8 | 63.50 | 64.12 |
| AdvLoRA+Nat | 2.8M | 47.1 | **71.8** | **81.9** | **66.93** | **47.5** | **71.2** | 79.9 | **66.20** | **66.57** |

# I ADAPTATION EFFICIENCY AND STORAGE COST

Table 16: Comparison on the training time and GPU memory.

| Method | Tuable Para. | Time | Memory |
|---|---|---|---|
| BLIP+FFT | 223M | 1.00 | 1.00 |
| BLIP+LoRA | 2.8M | 0.91 | 0.85 |
| BLIP+LP | 0.5M | 0.79 | 0.67 |
| Uniadapter | 19.5M | 0.93 | 0.77 |
| Aurora | 0.3M | 1.05 | 1.04 |
| AdvLoRA | 2.8M | 0.94 | 0.85 |

In this section, we conduct an analysis and comparison of the adaptation efficiency and storage cost associated with AdvLoRA. Table 16 illustrates the relative training GPU hours and GPU memory cost, where the time (or memory) of FFT is taken as one unit. The following conclusions can be drawn. 1) In terms of time overhead, AdvLoRA does not exhibit a pronounced advantage, but it outperforms Aurora and FFT. It is noteworthy that the adaptation process of models based on online weight decomposition, such as Aurora, requires more time than FFT. In contrast, AdvLoRA has a smaller time overhead due to the completion of only one offline clustering reparameterization and parameter alignment before adaptation. 2) In terms of memory overhead, AdvLoRA surpasses Aurora and FFT. Aurora again experiences a higher memory cost than FFT due to its heavier online decomposition. 3) Overall, AdvLoRA, without any additional constraints on training time and memory, can be considered an excellent adversarial adaptation method to enhance the adversarial robustness of VLMs. Note that AdvLoRA, LoRA, Uniadapter, and Aurora are essentially all parameter-efficient methods. The parameter-efficient techniques reduce the number of parameters to update, but they do not reduce the memory and time requirements during training by much since they still need to run the backward pass through the modelsSung et al. (2022). The main contribution of parameter-efficient methods is to reduce the costs of the model deploymentSung et al. (2022).

## J CASE STUDY

In this section, we conduct a case study on MSR-VTT, as illustrated in Figure 4. It can be observed that AdvLoRA achieves robust retrieval performance under adversarial attacks.

Video7168:
AdvLoRA retrieval result: He is playing with ball
Aurora retrieval result: Some people video conferencing as they watch a movie

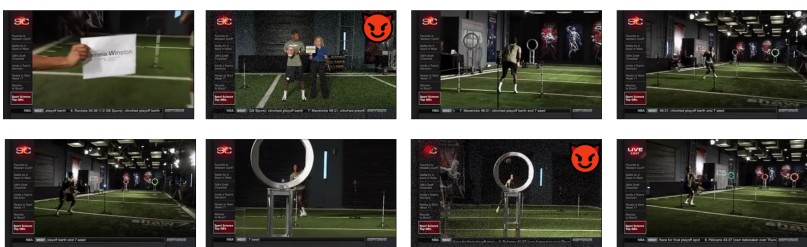

Video8915:
AdvLoRA retrieval result: Person cooking up somefood
Aurora retrieval result: A women preparing a duck to roast

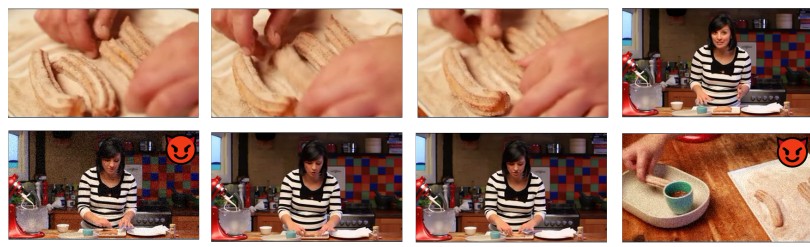

Video8128:
AdvLoRA retrieval result: A cartoon character prepares to ride a bicycle
Aurora retrieval result: People are walking down a street holding signs

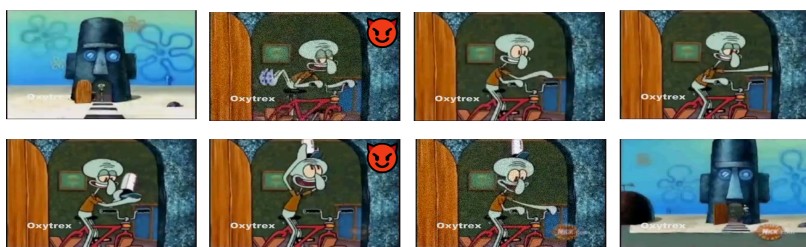

Figure 4: Case study of MSR-VTT. We sample and visualize eight frames from the videos. The frames with the devil denote that they are under the adversarial attacks. The first and second texts are the output of AdvLoRA and Aurora, respectively.

