# OpenReview forum: "AdvLoRA: Adversarial Low-Rank Adaptation of Vision-Language Models"
_ICLR.cc/2025/Conference — ICLR 2025 Conference Withdrawn Submission_

### Official Review · Reviewer_zUDk · 2024-10-19

**Soundness:** 4
**Presentation:** 4
**Contribution:** 2
**Rating:** 5
**Confidence:** 4

**Summary:**

This paper introduces a parameter-efficient method to enhance the adversarial robustness of VLMs. Traditional adaptation methods like full fine-tuning and LoRA are vulnerable to adversarial attacks, leading to performance drops. AdvLoRA improves robustness by utilizing low-rank adaptation, parameter clustering, and adaptive update strategies, reducing computational costs. Experiments show that AdvLoRA outperforms other methods, especially in adversarial scenarios.

**Strengths:**

1.	The writing is clear. The formulas are correct.
2.	The experiment is abundant and multi-dimensional.
3.	The research topic is important for VLM.

**Weaknesses:**

1.	While the method is effective, there is no analysis explaining the necessity of reparameterization.
2.	The rationale behind using clustering to establish a connection with the parameter in W is insufficiently analyzed.
3.	The justification for employing an adaptive update parameter is also lacking.

**Questions:**

Please see the weakness

---

### Official Review · Reviewer_NGsG · 2024-10-25

**Soundness:** 2
**Presentation:** 1
**Contribution:** 2
**Rating:** 3
**Confidence:** 4

**Summary:**

This  paper proposes a LoRA-based adversarial training method for visual language models. Unlike directly using LoRA, this method improves the efficiency and robustness of adversarial adaptation by designing a novel reparameterization method based on parameter clustering and parameter alignment. Through extensive experiments, the article demonstrates the effectiveness of AdvLora.

**Strengths:**

The paper provides a detailed introduction to the method, making it easy to understand.

It also conducts numerous experiments to demonstrate the effectiveness of the approach.

**Weaknesses:**

1. In terms of writing, the entire paper seems to not use the correct citation format; the ICLR template should utilize \citep. Therefore, a thorough review and verification of the paper are necessary to meet writing standards.
2. In lines 177-181, L has not used cross-referencing \ref.
3. It is a well-known fact that using adversarial samples for adversarial training can degrade model performance, and the introduction of Table 1 is not very clear regarding which model was trained.
4. If I am not mistaken, AdvLora seems to only improve the initialization of LoRA, which makes its contribution appear relatively small.
5. It is necessary to compare this method with other adversarial training approaches, such as RobustCLIP[1].

[1] Schlarmann, Christian, et al. "Robust clip: Unsupervised adversarial fine-tuning of vision embeddings for robust large vision-language models." arxiv preprint arxiv:2402.12336 (2024).

**Questions:**

See Weaknesses.
I would like to see the authors provide further clarification on the contributions of their work to confirm whether my understanding is correct.

---

### Official Review · Reviewer_72pu · 2024-11-02

**Soundness:** 4
**Presentation:** 4
**Contribution:** 3
**Rating:** 5
**Confidence:** 3

**Summary:**

The authors propose a parameter-efficient adversarial adaptation method called AdvLoRA, based on Low-Rank Adaptation. Initially, they investigate and reveal the intrinsic low-rank properties present in adversarial adaptation for vision-language models (VLMs). Unlike LoRA, AdvLoRA enhances the efficiency and robustness of adversarial adaptation through a novel reparameterization method that leverages parameter clustering and alignment. Additionally, an adaptive parameter update strategy is introduced to further enhance robustness. With these innovations, AdvLoRA addresses issues related to model security and excessive resource consumption. Extensive experiments demonstrate the effectiveness and efficiency of AdvLoRA.

**Strengths:**

1.	This paper presents AdvLoRA, a novel parameter-efficient adversarial adaptation method that improves the adversarial robustness of vision-language models (VLMs) through low-rank adaptation, representing an interesting avenue for research.
2.	The paper presents comparative results across some mainstream datasets.
3.	The method proposed in this paper is practical and applicable.

**Weaknesses:**

1.	The comparison between the proposed method and existing adversarial robustness techniques is insufficient, particularly regarding performance across different attack types.
2.	In the absence of an analysis of the proposed method's efficiency, clustering may be theoretically time-consuming.
3.	Ablation experiments should be a key component of the study, as it is crucial to evaluate the effectiveness of each module of the proposed method. The current content does not adequately demonstrate the method's effectiveness and lacks a detailed comparative analysis.
4.	The reparameterization method lacks theoretical support.

**Questions:**

1. Why does AB need to be aligned with W_0 ?

---

### Official Review · Reviewer_Peks · 2024-11-09

**Soundness:** 2
**Presentation:** 1
**Contribution:** 2
**Rating:** 3
**Confidence:** 5

**Summary:**

This paper focuses on the adversarial robustness of VLMs during PEFT. The authors improve the efficiency and robustness of adversarial adaptation by designing a reparametrizing method based on parameter clustering and parameter alignment.

**Strengths:**

This paper investigated an important problem and proved that the proposed ADVLORA can improve the adversarial robustness of BLIP-like VLMs in a parameter-efficient manner.

**Weaknesses:**

-	The novelty is very limited since the ADVLORA is proposed by combining adversarial training and LORA. Also, the proposed parameter clustering is not well-motivated.
-	The pipeline of ADVLORA is unclear. I hope that the authors could further clarify the purpose of Eq.8-12. Are they used for initialization or updated in each iteration?
-	How to choose the parameter $\alpha$, which is newly introduced compared to the original LORA.
-	The authors only investigate BLIP, whereas, there are many other VLMs, like CLIP.
-	The citation format should be revised. And there are many typos, such as “Eq. equation” in Algorithm1.

**Questions:**

- What is the purpose of Eq.8-12?
- How to choose the parameter $\alpha$?
- Does the ADVLORA work on other types of VLM?

---

### Note · Authors · 2024-11-13

I have read and agree with the venue's withdrawal policy on behalf of myself and my co-authors.